https://doi.org/10.1038/s41467-021-26004-5　　**OPEN**

# Bacterial chromosomal mobility via lateral transduction exceeds that of classical mobile genetic elements

Suzanne Humphrey [1], Alfred Fillol-Salom [1,2], Nuria Quiles-Puchalt [1,2], Rodrigo Ibarra-Chávez [1,3], Andreas F. Haag [1], John Chen[4] & José R. Penadés [1,2,5 ✉]

It is commonly assumed that the horizontal transfer of most bacterial chromosomal genes is limited, in contrast to the frequent transfer observed for typical mobile genetic elements. However, this view has been recently challenged by the discovery of lateral transduction in *Staphylococcus aureus*, where temperate phages can drive the transfer of large chromosomal regions at extremely high frequencies. Here, we analyse previously published as well as new datasets to compare horizontal gene transfer rates mediated by different mechanisms in *S. aureus* and *Salmonella enterica*. We find that the horizontal transfer of core chromosomal genes via lateral transduction can be more efficient than the transfer of classical mobile genetic elements via conjugation or generalized transduction. These results raise questions about our definition of mobile genetic elements, and the potential roles played by lateral transduction in bacterial evolution.

[1] Institute of Infection, Immunity and Inflammation, University of Glasgow, Glasgow G12 8TA, UK. [2] MRC Centre for Molecular Bacteriology and Infection, Imperial College London, London SW7 2AZ, UK. [3] Department of Biology, Section of Microbiology, University of Copenhagen, Universitetsparken 15, Bldg. 1, DK2100 Copenhagen, Denmark. [4] Department of Microbiology and Immunology, Yong Loo Lin School of Medicine, National University of Singapore, 5 Science Drive 2, Singapore, Singapore. [5] Departamento de Ciencias Biomédicas, Facultad de Ciencias de la Salud, Universidad CEU Cardenal Herrera, Valencia 46113, Spain. ✉email: j.penades@imperial.ac.uk

It has been classically assumed that the mobility of the bacterial chromosome is limited, in contraposition to that observed for the different members of the 'mobilome'. The mobilome is the umbrella term used to group all of the mobile genetic elements (MGEs) contained within a cell, with members typically falling within three main categories: plasmids, phages and phage-like elements, and transposable elements[1]. Many clinically and environmentally relevant bacteria harbour plasmids, transposable elements, prophages, and phage satellite elements for the efficient shuffling of genetic material between compatible cells. Conjugation is the main process by which plasmids are transferred intercellularly, and all plasmids can be broadly categorised as conjugative, mobilisable or non-transmissible[2]. Conjugative plasmids encode the mobilisation machinery that is necessary for their transfer to a new recipient cell[2]. In fact, the conjugative machinery has commonly been held to be the dominant mechanism by which horizontal gene transfer (HGT) is achieved[3]. Mobilisable plasmids are not capable of initiating conjugation on their own, but they can be mobilised by hijacking the machinery of conjugative plasmids[2]. Mobilisable and non-transmissible plasmids can be also mobilised in the lab via phage-mediated generalised transduction[4,5], although the impact of this in nature remains to be determined. Not just plasmids, but also integrative conjugative elements (ICEs) or conjugative transposons use conjugation for their self-transfer[6]. Plasmids, ICEs and transposons contribute to bacterial genomic diversity through the carriage of accessory and antibiotic-resistance genes[6,7]. ICEs and transposons can also affect gene expression through insertional inactivation or by altering promoter activity, depending on the manner in which they insert into the bacterial genome[1].

Phages and their counterparts, the phage-inducible chromosomal islands (PICIs)[8–10] are also key components of the mobilome and are important mediators of HGT. These elements readily transmit between host bacteria, where they can integrate into the bacterial chromosome and replicate passively during cell division. As new residents of a host cell, they can offer lysogenic immunity against phage superinfection or they can bestow important virulence phenotypes by introducing genes for toxins and colonisation factors[11,12]. In addition to mobilising their own DNA, phages can also mediate the exchange of bacterial DNA at very low frequencies through the processes of specialised and generalised transduction (ST and GT). Specialised transducing particles transmit restricted parts of the host chromosome, and they are formed by irregular prophage excision events that result in hybrid phage genomes that include bacterial DNA adjacent to phage attachment sites[13,14]. GT can transmit any bacterial DNA, and it occurs when host DNA is packaged into capsids at the expense of phage DNA to form transducing particles that inject their DNA cargo into a new host cell, where it can recombine into the host chromosome or exist as a plasmid[13,15]. GT is primarily mediated by *pac*-type phages, which package DNA by the headful mechanism. DNA packaging into transducing particles is initiated by the phage small terminase (TerS) at pseudo-*pac* (*ppac*) sites, which are sequences that resemble phage *pac* sites. These motifs are scattered throughout bacterial chromosomal and plasmid DNA and are recognised with varying frequencies that reflect their level of homology with bone fide *pac* sites[13].

While the mobilome seemed to be well defined, the broader concept of genetic mobility in bacteria is no longer well defined, as it has recently been upended by the discovery of the third and most powerful mode of phage-mediated DNA transfer: lateral transduction (LT)[16]. The LT mechanism begins with early in situ prophage replication, which creates multiple integrated prophages for genomic redundancy. Some prophages excise and enter the productive lytic cycle, while others serve as substrates for in situ DNA packaging from their embedded *pac* sites, which

are recognised far more efficiently than *ppac* sites (used in GT). When the first headful of DNA has been reached, the processive terminase enzyme continues to fill many more capsids with bacterial chromosome, which are subsequently transferred at high frequencies[16] (Fig. 1) and can then be integrated into the recipient genome by homologous recombination (HR). We propose here that when the full impact of this mechanism is considered, the classical dichotomy of portable MGEs and immobile chromosomes will no longer hold true because chromosomal genes can be mobilised at frequencies equal to or higher than elements normally regarded as mobile. To explore this idea, we looked to *Staphylococcus aureus* and *Salmonella* spp. as our reference organisms because their MGEs have been widely studied and they serve as model organisms for the mechanism of LT. We show that, via phage-mediated lateral transduction, the mobility of core genes in bacterial chromosomes can exceed that of elements classically considered to be mobile.

## Results and discussion

**Comparison of horizontal transfer rates**. A comparison of the many *S. aureus* mobilome elements revealed striking differences in their horizontal transfer frequencies. The staphylococcal pathogenicity islands (SaPIs), the prototypical members of the PICI family[8–10], mobilised themselves at rates at least 1000-fold higher than any plasmid or transposable element (Fig. 2a, Table 1), showing that PICIs are exquisitely adapted for maximising their own transfer between host cells. Notably, SaPI1 transduced between cells at very high frequency, with the equivalent of at least two successful transfer events (TE) occurring from each bacterial donor cell. Similarly, phage φSLT exhibited an efficient lysogenisation rate (integration into the recipient strain) of approximately 1 lysogen produced from 130 donor cells (Fig. 2a, Table 1). In order to extend our analysis to include more staphylococcal phages, we also tested the transfer rates of phages 80α and φ11 (Fig. 2a, Tables 1 and 2), which achieved lysogenisation rates in the range of approximately 1 lysogen from 62 donor cells and 1 lysogen from 2 donor cells, respectively, indicating the efficiency of transfer by these elements and highlighting the role played by phages in generating extensive bacterial diversity in nature. Note that we have compared lysogenisation rates here and not plaque formation because the latter does not provide any extra gene to the recipient cells, which incidentally will die at the end of the infective process. Interestingly, despite variance between the different conjugative and mobilisable plasmids, the transfer frequencies were considerably lower than those of the phages and SaPIs for all of the plasmids analysed but were higher than those of the conjugative transposons (Fig. 2a, Table 1). These data highlight that while all of these elements are considered to be mobile, there can be extreme differences in their relative transfer efficiencies. As expected, based on the classical dichotomy of MGEs vs the immobile chromosome, 80α-mediated GT transferred the chromosomally-encoded cadmium-resistance (Cad$^R$) markers at significantly lower frequencies than the transfer of the phage itself. Unexpectedly, the transfer frequencies of the Cad$^R$ markers by GT were comparable to the range reported for the conjugative plasmids and were higher than those reported for the conjugative transposons, suggesting that DNA mobilisation by GT may not be as rare an event as typically assumed. Furthermore, this observation was reinforced when we compared the transfer rate of the 'non-mobilisable' plasmid pJP2511 via 80α or φ11-mediated generalised transduction (Table 2), with pJP2511 GT transfer efficiencies exceeding those reported for the classically mobile conjugative and mobilisable plasmids analysed (Fig. 2a, Table 1), lending weight to the importance of the role of GT in driving lineage

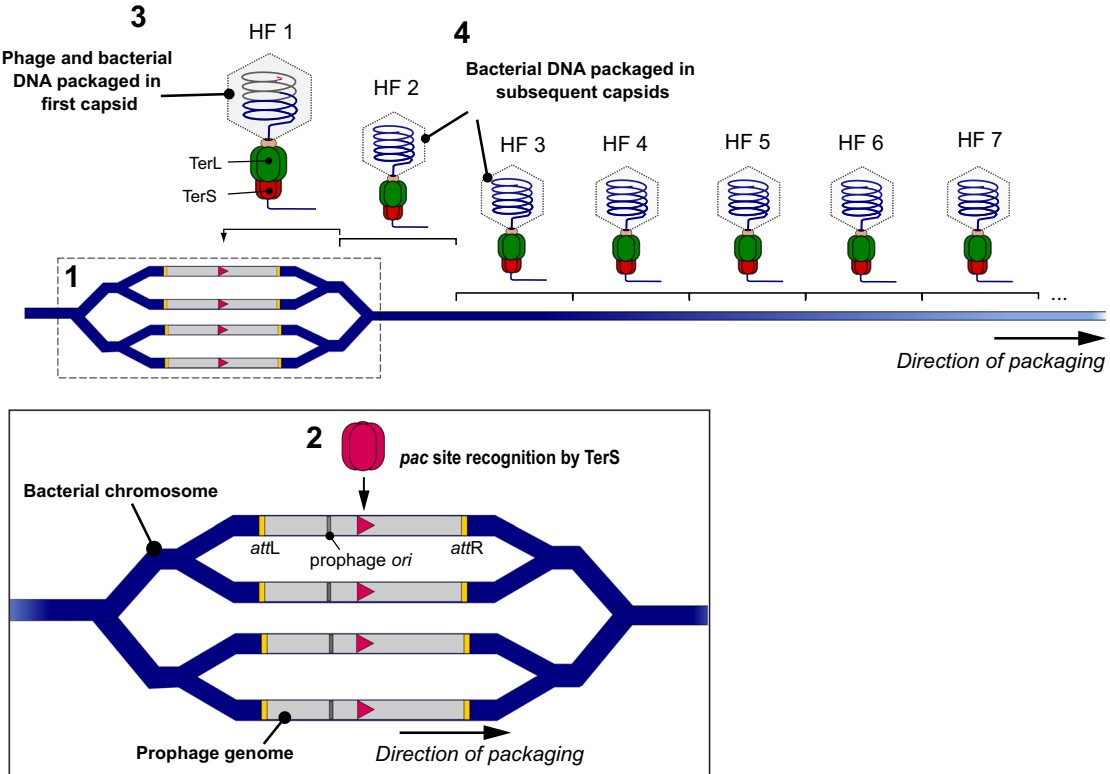

**Fig. 1 Lateral transduction of chromosomal DNA.** LT is initiated during the early stages of prophage activation with the prophage remaining integrated in the bacterial chromosome. The mechanism commences with bidirectional in situ replication of the integrated prophage from the phage *ori*, generating multiple copies of the integrated phage and surrounding bacterial chromosome [1]. Some prophages subsequently excise from the chromosome to generate progeny via the lytic cycle, but in those that remain integrated, the phage small terminase subunit (TerS) recognises the embedded *pac* site (pink triangle) within the prophage sequence, forming a complex for delivery of the DNA to the large terminase (TerL) subunit [2]. The TerS:DNA complex associates with the large terminase (TerL) subunit, which cleaves and translocates the DNA into available phage capsids until capacity (one headful) is reached, with the initial capsid containing a mixture of phage and chromosomal DNA [3]. When the capsid is filled, the DNA is cleaved once more and the Terminase:DNA complex associates with a new empty capsid to resume the packaging process, generating many processive headfuls containing bacterial chromosomes for subsequent transduction [4].

diversification via horizontal gene transfer. Critically, 80α-mediated LT transferred the same Cad$^R$ markers at substantially higher efficiencies than those reported for the other elements (Fig. 2a, Table 1), excepting phage and SaPI transfer, with a transfer efficiency of the first phage headful achieving a similar level of efficiency ($5.18 \times 10^{-2}$ events/donor) to that observed for the phage itself ($1.69 \times 10^{-2}$ events/donor).

Since the previous results completely challenged our concept of DNA mobility, we decided to include an additional species to validate this paradigm change. A parallel analysis of mobilome components in *Salmonella* revealed that phage P22 is exquisitely well-adapted for transfer between cells, with approximately 1.57 new lysogens produced for every donor cell-induced (Fig. 3a, Table 3). Interestingly, this analysis also revealed considerably more variation in the transfer frequency rates of *Salmonella* conjugative plasmids than was observed for their Gram-positive counterparts (Fig. 3a, Table 3). Transfer of the *Salmonella* conjugative plasmid pOU1114 appeared to be highly efficient, with 1 transconjugant produced from approximately 23 donor cells, making it the second most mobile element examined in the *Salmonella* panel. In contrast, however, the remaining *Salmonella* conjugative plasmids had transfer frequencies in the range $10^{-8}$–$10^{-4}$ TE per donor, which was similar to the range of transfer frequencies observed for the two *Salmonella* ICEs analysed (Fig. 3a, Table 3). Notably, similarly to that observed in *S. aureus*, P22-mediated GT of chromosomally encoded tetracycline-resistance (*tet*$^R$) markers had transfer frequencies

greater than or equivalent to the range reported for the conjugative plasmids and ICEs, excepting pOU1114. Importantly, the transfer frequency of *tet*$^R$ markers by P22-mediated LT was again higher than that observed for all except one of the classical MGEs analysed (Fig. 3a, Table 3), indicating that LT in this species, and to a lesser extent GT: (i) challenge the concept of what represents an MGE, and (ii) represent important pathways for efficient genetic exchange in both Gram-positive and Gram-negative organisms.

**Cargo carriage by MGEs: benefit or burden?** Having established that chromosomal genes can be mobilised at higher frequencies than many (if not most) classical MGEs, we then considered the relative contribution of each element to gene exchange in terms of the quantity of potential 'useful' DNA transferred (the *cargo capacity*). Importantly, though phages and SaPIs transduce at extremely high frequencies, they typically carry four or fewer genes of overt benefit to their host bacterium[8,12]. Plasmids vary widely in terms of size and hence their cargo capacity also varies considerably, and though clearly larger conjugative plasmids have substantially higher potential cargo capacity than that of SaPIs and phages, they are severely limited by having to encode all of the necessary genes for conjugative transfer without exceeding the metabolic capacity of their host cell. Arguably, owing to the fact that LT and GT package chromosomal DNA into empty phage capsids for transfer without requiring any capacity for replicative function, the cargo capacity of LT or GT virions is maximised.

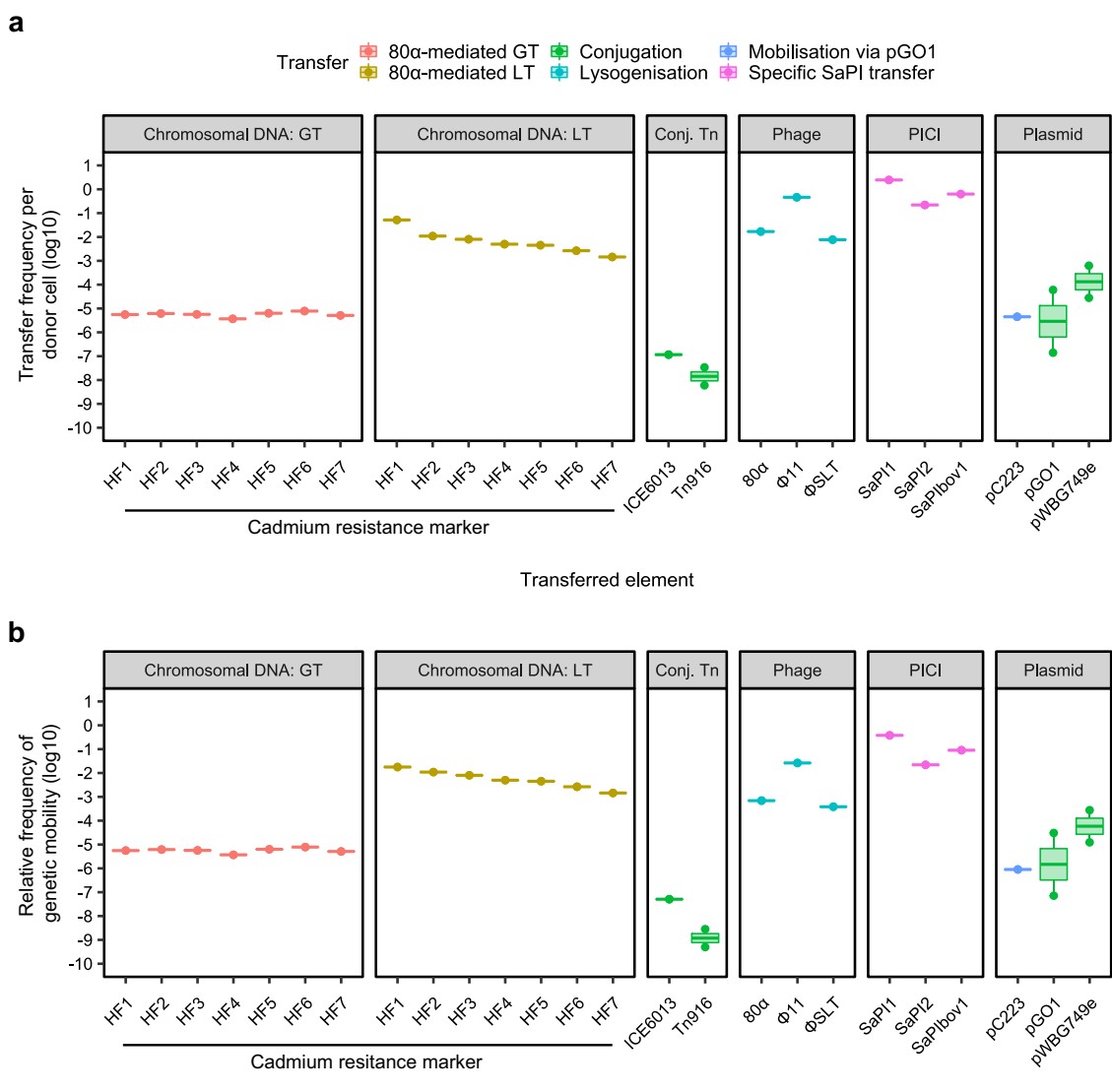

**Fig. 2 Genetic mobility via different mechanisms in *S. aureus*. a** Transfer frequencies per donor cell and **b** relative frequency of genetic mobility (defined as transfer frequency × cargo capacity) of different genetic elements by transfer type. Transfer of chromosomal DNA by phage 80α via generalised (GT) or lateral (LT) transduction of a cadmium resistance marker located at defined distances from the phage attachment site, conjugative transposons (Conj. Tn), phages, PICIs and plasmids. Data were extracted from literature or acquired through experimentation (see Table 1 for details).

Indeed, when the overall genetic mobility rate (transfer frequency × cargo capacity) is compared for each element in *S. aureus* (Fig. 2b, Table 1), it is striking that in our analysis the efficiency of LT exceeds that of all other elements excepting only the SaPIs and φ11, which exhibit incredibly efficient transfer rates of accessory genes between donor and recipient strains despite their relatively low cargo capacities. In *Salmonella*, the rate of overall genetic mobility for LT exceeded that of the conjugative plasmids pSLT and pS3, and conjugative elements ICESb1 and SGI3 (Fig. 3b, Table 3), indicating that transfer of the chromosome is at least equal to, if not more efficient than, transfer of elements classically considered to be 'mobile'.

Importantly, in the case of lateral transduction by phage 80α each capsid can accommodate approximately 45–46 kb of chromosomal DNA, with highly efficient packaging of DNA occurring up to seven or more headfuls away from the phage integration site in the direction of packaging, implicating the potential for a single prophage to transfer up to 315 kb of bacterial DNA by LT[16]. Similarly, P22-mediated lateral transduction has been demonstrated up to 12 headfuls away from the phage integration site[17], with each individual P22 capsid being capable of accommodating ~43–44 kb of chromosomal DNA, indicating that P22-mediated LT has the potential to efficiently mobilise at least 516 kb of bacterial DNA from a single prophage. This alone is impressive in its potential for gene exchange, but the prospective magnitude for chromosomal mobilisation is brought sharply into focus when considering the chromosomal architecture of phage attachment sites in *S. aureus* and *S. enterica*.

**Table 1 Genetic mobility via different mechanisms in S. aureus.**

| Mobilome component | Accession | DNA transferred | HGT mechanism | Transfer frequency (TE/donor CFU)[a] | Source for transfer frequency data | Cargo capacity rate[b] | Relative frequency of genetic mobility[c] |
|---|---|---|---|---|---|---|---|
| *Plasmids* | | | | | | | |
| pGO1 | NC_012547 | Plasmid | Conjugation | $6.0 \times 10^{-5}$ to $1.4 \times 10^{-7}$ | 42,43 | 0.51 (28/55) | $3.05 \times 10^{-5}$ to $7.13 \times 10^{-8}$ |
| pAM387 | | Plasmid | Conjugation | $1.1 \times 10^{-6}$ | 44 | ND | ND |
| pWBG749e | | Plasmid | Conjugation | $6.25 \times 10^{-4}$ to $2.8 \times 10^{-5}$ | 45 | 0.44 (23/52) | $2.76 \times 10^{-4}$ to $1.24 \times 10^{-5}$ |
| pC223 | NC_005243 | Plasmid | Mobilisation via pGO1 | $4.5 \times 10^{-6}$ | 42,46 | 0.20 (1/5) | $9.00 \times 10^{-7}$ |
| *Phage and PICI elements* | | | | | | | |
| φSLT | AB045978.2 | Phage | Lysogenisation | $7.69 \times 10^{-3}$ | 47 | 0.05 (3/60) | $3.85 \times 10^{-4}$ |
| 80α | NC_009526 | Phage | Lysogenisation | $1.69 \times 10^{-2}$ | This work | 0.04 (3/73) | $6.95 \times 10^{-4}$ |
| φ11 | AF424781 | Phage | Lysogenisation | $4.62 \times 10^{-1}$ | This work | 0.06 (3/53) | $2.62 \times 10^{-2}$ |
| SaPI | U93688.2 | PICI | Specific SaPI transfer | 2.46 | 26 | 0.15 (4/26) | $3.78 \times 10^{-1}$ |
| SaPIbov1 | AF217235.1 | PICI | Specific SaPI transfer | $6.31 \times 10^{-1}$ | 26 | 0.14 (3/21) | $9.01 \times 10^{-2}$ |
| SaPI2 | EF010993 | PICI | Specific SaPI transfer | $2.20 \times 10^{-1}$ | 26 | 0.04 (2/24) | $2.20 \times 10^{-2}$ |
| *Transposons—ICEs* | | | | | | | |
| ICE6013 | PRJNA360134 (strain DAR6247) | Conjugative Tn | Conjugation | $1.16 \times 10^{-7}$ | 48 | 0.44 (7/16) | $5.08 \times 10^{-8}$ |
| Tn916 | U09422 | Conjugative Tn | Conjugation | $6.0 \times 10^{-9}$ to $3.4 \times 10^{-8}$ | 49 | 0.08 (2/24) | $5.00 \times 10^{-10}$ to $6.25 \times 10^{-11}$ |
| *Chromosomal transfer* | | | | | | | |
| *Generalised transduction* | | | | | | | |
| CadR marker HF[d] | NC_007795.1 | Chromosomal DNA | 80α-mediated GT | $5.58 \times 10^{-6}$ | 16 | 1.00 (47/47)[e] | $5.58 \times 10^{-6}$ |
| CadR marker HF3 | | Chromosomal DNA | 80α-mediated GT | $5.69 \times 10^{-6}$ | 16 | 1.00 (39/39) | $5.69 \times 10^{-6}$ |
| CadR marker HF5 | | Chromosomal DNA | 80α-mediated GT | $6.31 \times 10^{-6}$ | 16 | 1.00 (40/40) | $6.31 \times 10^{-6}$ |
| CadR marker HF7 | | Chromosomal DNA | 80α-mediated GT | $5.12 \times 10^{-6}$ | 16 | 1.00 (41.5/41.5) | $5.12 \times 10^{-6}$ |
| *Lateral transduction* | | | | | | | |
| CadR marker HF1 | NC_007795.1 | Chromosomal DNA | 80α-mediated LT | $5.18 \times 10^{-2}$ | 16 | 0.34 (17.5/51)[f] | $1.78 \times 10^{-2}$ |
| CadR marker HF3 | | Chromosomal DNA | 80α-mediated LT | $8.00 \times 10^{-3}$ | 16 | 1.00 (39/39) | $8.00 \times 10^{-3}$ |
| CadR marker HF5 | | Chromosomal DNA | 80α-mediated LT | $4.51 \times 10^{-3}$ | 16 | 1.00 (40/40) | $4.51 \times 10^{-3}$ |
| CadR marker HF7 | | Chromosomal DNA | 80α-mediated LT | $1.45 \times 10^{-3}$ | 16 | 1.00 (41.5/41.5) | $1.45 \times 10^{-3}$ |

ND not determined.

Source data are provided as a Source Data file.

[a] In order to enable comparisons between conjugation and the other modes of horizontal gene transfer, transfer frequency of phage- and PICI-mediated DNA transfer was re-analysed as transfer events (TE) per bacterial donor cell at the time of prophage induction: TE per donor cell = Transductant Units (TrU) per ml/$6.5 \times 10^7$ CFU per ml. TrU per ml data were obtained from the sources indicated. Lysogens were induced at $OD_{540}$ 0.15, which is equivalent to $6.5 \times 10^7$ CFU per ml in the donor population.

[b] Cargo capacity rate = number of accessory ORFs utilisable by the host cell (e.g. virulence factors and AMR genes)/total ORFs contained within the mobilised DNA sequence. Bracketed values indicate the number of accessory ORFs for each element where sequence data was available for analysis. For the purposes of this analysis, these staphylococcal phages are proposed to carry three ORFs with lysogenic conversion effects (e.g. toxins or phage-resistance mechanisms), while SaPIs carry 2-4 ORFs conferring enhanced virulence characteristics for the lysogenic host.

[c] Relative frequency of genetic mobility = transfer frequency × cargo capacity rate.

[d] HF, phage headful (~45 kb); numbers denote the distance of each cadmium marker from the phage chromosomal attachment site in terms of headful units in the direction of phage packaging.

[e] Estimation of ORFs packaged in HF1 during generalised transduction if packaging terminates in the same location as for HF1 during lateral transduction. No phage genes are expected to be transduced during generalised transduction of the DNA sequence containing the cadmium resistance marker, so 100% of the transferred sequence is available for recombination and utilisation by the recipient cell.

[f] The total number of accessory ORFs utilisable by the recipient host cell is only a proportion of the total sequence transferred by HF1 because part of the phage genome is also packaged in the first headful during lateral transduction.

**Table 2 Transfer rates of phages, plasmids and chromosomal markers.**

| Strain | Phage titre (PFU per ml) Mean (±SD) | Phage transfer (lysogenisation) | | Chromosome transfer (phage-mediated lateral transduction) | | Plasmid pJP2511 transfer (phage-mediated generalised transduction) | |
|---|---|---|---|---|---|---|---|
| | | Mean transduction titre (TrU/ml; ±SD) | Transfer frequency (TE/donor)[a] | Mean transduction titre (TrU/ml ±SD) | Transfer frequency (TE/donor) | Mean transduction titre (TrU/ml; ±SD) | Transfer frequency (TE/donor) |
| *Staphylococcus aureus* | | | | | | | |
| JP20844 (φ11) | $6.00 \times 10^8$ ($\pm 1.83 \times 10^8$) | $3.00 \times 10^7$ ($\pm 2.31 \times 10^7$) | $4.62 \times 10^{-1}$ | $5.00 \times 10^5$ ($\pm 1.15 \times 10^4$) | $7.69 \times 10^{-3}$ | $2.00 \times 10^5$ ($\pm 1.15 \times 10^5$) | $3.08 \times 10^{-3}$ |
| JP20846 (80α) | $1.15 \times 10^9$ ($\pm 7.23 \times 10^8$) | $1.10 \times 10^6$ ($\pm 6.16 \times 10^5$) | $1.69 \times 10^{-2}$ | $3.25 \times 10^5$ ($\pm 2.63 \times 10^5$) | $5.00 \times 10^{-3}$ | $5.50 \times 10^4$ ($\pm 5.77 \times 10^3$) | $8.46 \times 10^{-4}$ |
| *Salmonella* Typhimurium | | | | | | | |
| JP22210 (P22) | $2.03 \times 10^8$ ($\pm 1.10 \times 10^8$) | $1.57 \times 10^8$ ($\pm 8.33 \times 10^7$) | 1.567 | ND | ND | ND | ND |

ND not determined.
Source data are provided as a Source Data file.
Data are the mean values of four (φ11 and 80α) or three (P22) independent experiments.
[a]TE, transfer events. TE per donor cell = transductant units (TrU) per ml/6.5 × 10⁷ CFU per ml for *S. aureus* phages, or transductant units (TrU) per ml/1.0 × 10⁸ CFU per ml for *S.* Typhimurium phage P22.

The distribution and orientation of ten commonly used *att*B sites in the *S. aureus* chromosome are arranged in such a way that a poly-lysogenic strain could potentially mobilise up to 1.7 Mb of DNA (approximately 60% of the *S. aureus* chromosome) via LT in a single lytic event, presenting an astonishing potential for widespread horizontal gene exchange (Fig. 4a). A similar observation can be made for *S.* Typhimurium strain LT2, with poly-lysogeny offering the potential for transfer of up to 3.5 Mb of the bacterial chromosome in a single step (Fig. 4b). Together, these results indicate that phages offer highly efficient platforms for the transfer of significant portions of chromosomal DNA at rates higher than those of many other MGEs, indicating that through LT the bacterial chromosome becomes more mobile than many classical MGEs.

**Exploring the contribution of LT to virulence evolution**. The scope and efficiency conferred by LT for the intercellular transfer of chromosomal genes raise questions over whether the bacterial chromosome itself could qualify as an MGE. MGEs are typically considered to be 'selfish' elements, promoting their own dissemination at the expense of other genetic elements in the cell. If we are to assume that parts of the bacterial chromosome itself behave akin to classical MGEs, we must also consider whether certain chromosomal loci also exhibit selfish interests for mediating their intercellular transmission. Classical MGEs are known to carry distinct genetic cargos that are enriched in genes that provide locally adaptive traits like toxin production, resistance, and virulence[18,19], relative to the chromosome (as a whole). This is because those genes benefit (selfishly) from being able to move into new genetic backgrounds[20,21]. If parts of the chromosome are indeed hypermobile, theory suggests they should resemble MGEs in terms of gene content. In such cases, it would be expected that prophage-adjacent regions of the chromosome should indicate some enrichment in genes conferring a competitive advantage to the host cell, such as genes involved in virulence or environmental adaptation, relative to the rest of the chromosome. Examination of the chromosomal architecture of *S. aureus* and *S.* Typhimurium provides compelling circumstantial evidence that this may indeed be the case for several pathogenicity-associated genetic regions in each species (Table 4).

The Staphylococcal genomic islands (GI) vSaα, vSaβ, and vSaγ are pathogenicity islands of DNA located within the *S. aureus* chromosome that encode a wide variety of genes involved in Staphylococcal virulence, including enterotoxins, serine proteases and leukotoxins[22]. Carriage of vSaα and vSaβ occurs widely among *S. aureus* strains[23], with many strains also harbouring the third GI, vSaγ[24]. The absence of vSaγ from some strains strongly suggests that these elements have been obtained by *S. aureus* via HGT, with vSaγ being the most recent acquisition. Until recently, however, no mechanism had been ascribed that was able to satisfactorily explain how this widely spread TE could occur in nature. We believe that the discovery of LT provides an elegant mechanism explaining how such transfers can be facilitated at high frequency. Our reasoning for this is twofold: firstly, it is striking that each of the GIs is located in the *S. aureus* chromosome such that it is close to a prophage integration site (*att*B) in the direction of phage packaging, permitting compatibility with packaging and transfer via phage-mediated LT: vSaα is located downstream of the Sa6 prophage *att*B, compatible with packaging into HF3; vSaγ is located close (within two headfuls) to Sa7 *att*B, and is also compatible with packaging from Sa4 (~HF4-5) and Sa1 (~HF7) prophage attachment sites; while vSaβ is located close to attachment sites Sa8 (~HF1-2), Sa5 (~HF2-3), as well as Sa3 (~HF5-6)[16]. Secondly, two recent studies have demonstrated a role for staphylococcal prophages in the transduction of the GI[24,25].

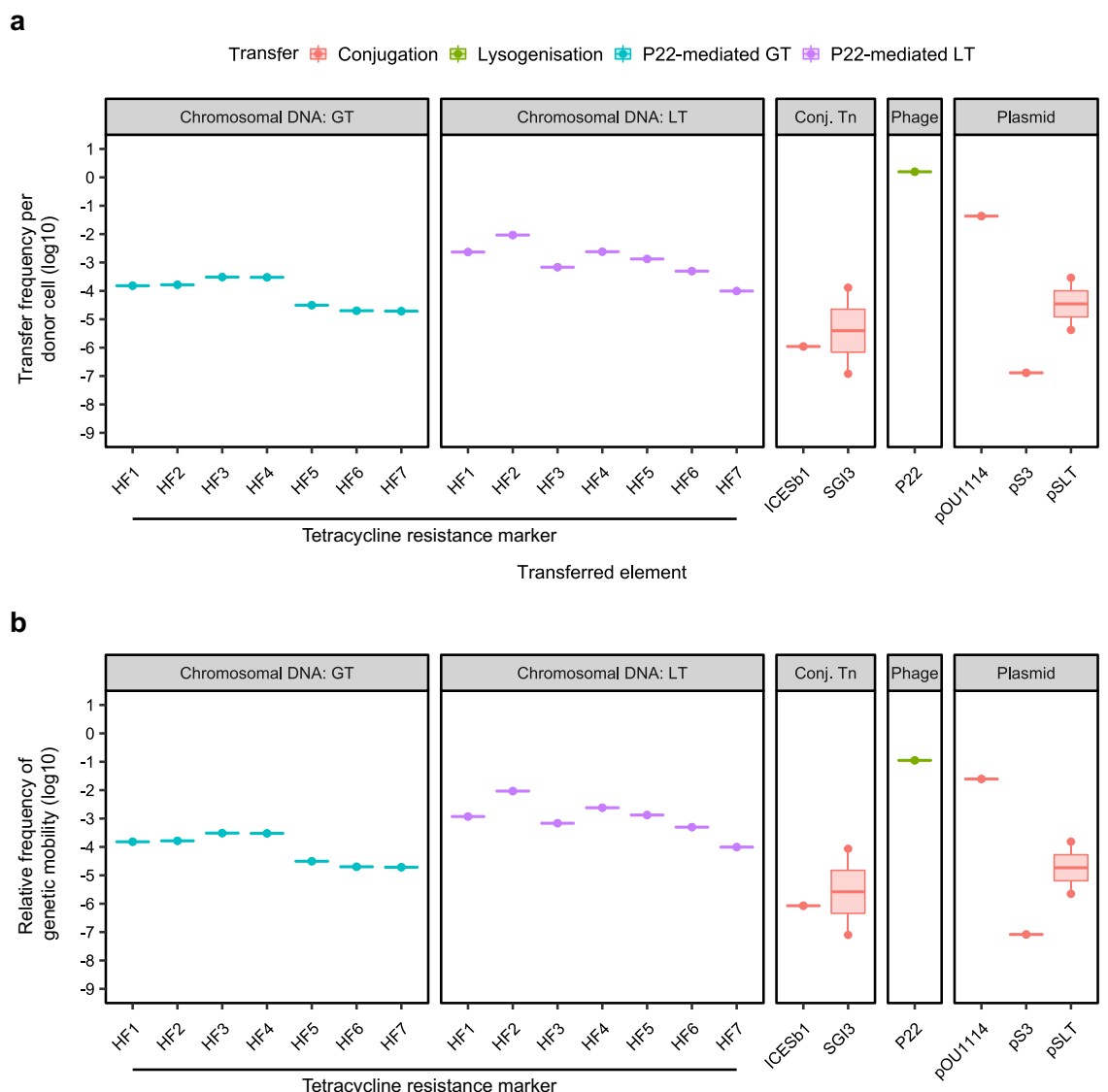

**Fig. 3 Genetic mobility via different mechanisms in *Salmonella* spp. a** Transfer frequencies per donor cell and **b** relative frequency of genetic mobility (defined as transfer frequency × cargo capacity) of different genetic elements by transfer type. Transfer of chromosomal DNA by phage P22 via generalised (GT) or lateral (LT) transduction of a tetracycline resistance marker located at defined distances from the phage attachment site, conjugative transposons (Conj. Tn), phages and plasmids. Data were extracted from literature or acquired through experimentation (see Table 3 for details).

Notably, the first of these studies showed transduction of vSaβ mediated by a prophage integrated at the Sa8 attachment site adjacent to the island. In this study, the authors describe phage-mediated transduction of the island via a complex double transduction event, where they speculate that overlapping sections of DNA from multiple transducing particles result in the transfer of vSaβ via HR[25]. It is interesting, however, that the authors describe transducing particles containing phage-vSaβ hybrid sections of DNA, a feature that is now known to be observed in HF1 of LT particles, suggesting the potential involvement of LT in the process of vSaβ mobilisation. In addition, the authors found a positive correlation between the presence of both vSaβ and a phage localised adjacent to the island among a panel of bovine *S. aureus* isolates, further suggesting a potential role for phage-mediated LT in the dissemination of this GI.

Similarly, it is interesting to speculate that LT may also have played a role in the evolution of the SaPIs in *S. aureus*. The organisation of the five SaPI chromosomal attachment sites

described for *S. aureus* are orientated such that they are usually adjacent to a prophage integration site in the direction of phage packaging, permitting SaPI packaging and transfer via phage-mediated LT. Indeed, SaPIs integrated at SaPI Type I and II attachment sites could readily be mobilised via LT within HF2 or HF3, respectively, of a prophage integrated at Sa6 *attB*. Similarly, SaPI integration at SaPI Type III or IV *attB* sites would permit packaging of the island within the first (Type III) or second (Type IV) headful of a prophage integrated at Sa9 *attB*, or even from Sa12 *attB* (within ~HF6)[16]. Our analysis has shown that SaPI-specific transfer is highly efficient between cells at the expense of the helper phage, however, this level of transfer is entirely reliant on de-repression of the SaPI by its helper phage, allowing it to replicate and hijack the phage machinery[26]. In evolutionary terms, insertion of SaPI elements into the chromosome adjacent to phage attachment sites to permit exploitation of LT would provide a further 'selfish' mechanism for SaPI transfer by non-helper phages, promoting survival of the elements under

**Table 3 Genetic mobility via different mechanisms in *Salmonella* spp.**

| | Accession | DNA transferred | HGT mechanism | Transfer frequency (TE/donor CFU)[a] | Source for transfer frequency data | Cargo capacity rate[b] | Relative frequency of genetic mobility[c] |
|---|---|---|---|---|---|---|---|
| *Mobilome component* | | | | | | | |
| Plasmids | | | | | | | |
| pSLT | CP001362 | Plasmid | Conjugation | $4.20 \times 10^{-6}$ to $2.90 \times 10^{-4}$ | 50 | 0.53 (54/102) | $2.22 \times 10^{-6}$ to $1.54 \times 10^{-4}$ |
| pS3 | DQ115387 | Plasmid | Conjugation | $1.30 \times 10^{-7}$ | 51 | 0.63 (52/82) | $8.24 \times 10^{-8}$ |
| pOU1114 | | Plasmid | Conjugation | $4.30 \times 10^{-2}$ | 52 | 0.57 (27/47) | $2.47 \times 10^{-2}$ |
| pESI | | Plasmid | Conjugation | $4.00 \times 10^{-6}$ | 53 | ND | ND |
| pWWO12 | CP022169 | Plasmid | Conjugation | $1.20 \times 10^{-6}$ | 54 | ND | ND |
| Phage | | | | | | | |
| P22 | | Phage | Lysogenisation | 1.57 | This work | 0.07 (5/70) | $1.12 \times 10^{-1}$ |
| Transposable elements—ICEs | | | | | | | |
| ICESb1 | FN298494.1 | Conjugative Tn | Conjugation | $1.10 \times 10^{-6}$ | 55 | 0.77 (81/105) | $8.49 \times 10^{-7}$ |
| SGI3 | | Conjugative Tn | Conjugation | $1.20 \times 10^{-7}$ to $1.30 \times 10^{-4}$ | 56 | 0.66 (57/86) | $7.95 \times 10^{-8}$ to $8.62 \times 10^{-5}$ |
| *Chromosomal transfer* | | | | | | | |
| Generalised transduction | | | | | | | |
| tetA marker HF1[d] | AE006468.2 | Chromosomal DNA | P22-mediated GT | $1.52 \times 10^{-4}$ | 17 | 1.00 (41/41)[e] | $1.52 \times 10^{-4}$ |
| tetA marker HF2 | | Chromosomal DNA | P22-mediated GT | $1.64 \times 10^{-4}$ | 17 | 1.00 (33.5/33.5) | $1.64 \times 10^{-4}$ |
| tetA marker HF3 | | Chromosomal DNA | P22-mediated GT | $3.07 \times 10^{-4}$ | 17 | 1.00 (44/44) | $3.07 \times 10^{-4}$ |
| tetA marker HF4 | | Chromosomal DNA | P22-mediated GT | $3.00 \times 10^{-4}$ | 17 | 1.00 (37/37) | $3.00 \times 10^{-4}$ |
| tetA marker HF5 | | Chromosomal DNA | P22-mediated GT | $3.13 \times 10^{-5}$ | 17 | 1.00 (42/42) | $3.13 \times 10^{-5}$ |
| tetA marker HF6 | | Chromosomal DNA | P22-mediated GT | $2.00 \times 10^{-5}$ | 17 | 1.00 (43/43) | $2.00 \times 10^{-5}$ |
| tetA marker HF7 | | Chromosomal DNA | P22-mediated GT | $1.93 \times 10^{-5}$ | 17 | 1.00 (43/43) | $1.93 \times 10^{-5}$ |
| Lateral transduction | | | | | | | |
| tetA marker HF1 | AE006468.2 | Chromosomal DNA | P22-mediated LT | $2.33 \times 10^{-3}$ | 17 | 0.51 (21.5/42.5) | $1.18 \times 10^{-3}$ |
| tetA marker HF2 | | Chromosomal DNA | P22-mediated LT | $9.27 \times 10^{-3}$ | 17 | 1.00 (33.5/33.5) | $9.27 \times 10^{-3}$ |
| tetA marker HF3 | | Chromosomal DNA | P22-mediated LT | $6.83 \times 10^{-4}$ | 17 | 1.00 (44/44) | $6.83 \times 10^{-4}$ |
| tetA marker HF4 | | Chromosomal DNA | P22-mediated LT | $2.40 \times 10^{-3}$ | 17 | 1.00 (37/37) | $2.40 \times 10^{-3}$ |
| tetA marker HF5 | | Chromosomal DNA | P22-mediated LT | $1.33 \times 10^{-3}$ | 17 | 1.00 (42/42) | $1.33 \times 10^{-3}$ |
| tetA marker HF6 | | Chromosomal DNA | P22-mediated LT | $4.97 \times 10^{-4}$ | 17 | 1.00 (43/43) | $4.97 \times 10^{-4}$ |
| tetA marker HF7 | | Chromosomal DNA | P22-mediated LT | $9.87 \times 10^{-5}$ | 17 | 1.00 (43/43) | $9.87 \times 10^{-5}$ |

ND not determined.

Source data are provided as a Source Data file.

[a]In order to enable comparisons between conjugation and the other modes of horizontal gene transfer, transfer frequency of phage-mediated DNA transfer was analysed as transfer events (TE) per bacterial donor cell at the time of prophage induction: TE per donor cell = Transductant Units (TrU) per ml/1.0 × 10$^8$ CFU per ml. TrU per ml data was obtained from the sources indicated. Lysogens were induced at OD$_{600}$ 0.2, which is equivalent to 1.0 × 10$^8$ CFU per ml in the donor population.

[b]Cargo capacity rate = Number of accessory ORFs utilisable by the host cell (e.g., virulence factors, AMR genes and HPs)/total ORFs contained within the mobilised DNA sequence. Bracketed values indicate the number of accessories ORFs/total ORFs for each element where sequence data was available for analysis. Phage P22 is proposed to carry five ORFs with lysogenic conversion effects: *sieAB* and *gtrABC*.

[c]Relative frequency of genetic mobility = transfer frequency × cargo capacity rate.

[d]HF, phage headful (43.8 kb); numbers denote the distance of each tetracycline-resistance marker from the phage chromosomal attachment site in terms of headful units in the direction of phage packaging.

[e]Estimation of ORFs packaged in HF1 during generalised transduction if packaging terminates in the same location as for HF1 during lateral transduction. No phage genes are expected to be transduced during generalised transduction of the DNA sequence containing the tetracycline-resistance marker, so 100% of the transferred sequence is available for recombination and utilisation by the recipient cell.

[f]The total number of accessory ORFs utilisable by the recipient host cell is only a proportion of the total sequence transferred by HF1 because part of the phage genome is also packaged in the first headful during lateral transduction.

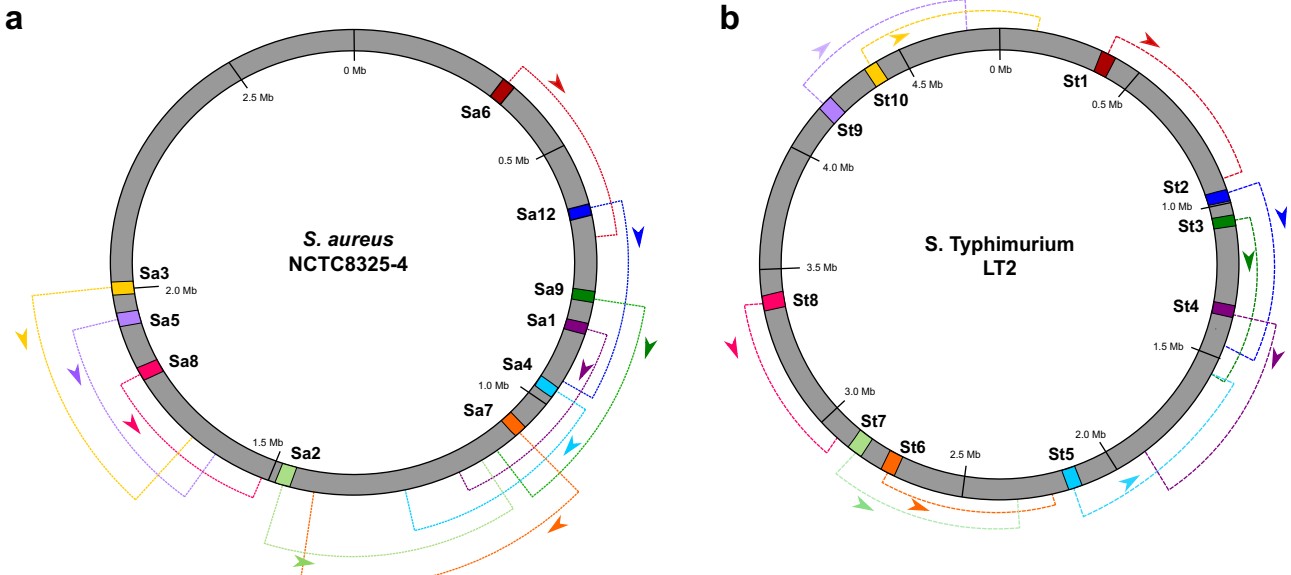

**Fig. 4 Potential for extensive DNA transfer via lateral transduction is dictated by the distribution of prophage attachment sites in the bacterial chromosome.** Map of the chromosomes of *S. aureus* NCTC8325-4 (**a**) and *S.* Typhimurium LT2 (**b**) indicating the ten potential chromosomal attachment (*attB*) sites available for prophage integration. The packaging direction of an integrated prophage from each *attB* site is indicated by its corresponding colour-coded arrow, with the dashed regions representing a distance of approximately seven headfuls (one headful = ~45 kb for 80α) in *S. aureus*, or 12 headfuls (one headful = ~43 kb for P22) in *S.* Typhimurium from the *attB* site, which is the minimum distance known to be packaged by LT for each phage. **a** The distribution and directionality of *attB* sites in the *S. aureus* NCTC8325-4 chromosome indicates the potential for transfer of the entire region between ~0.31 and 2.0 Mb from a poly-lysogenic background, representing approximately 58% of the bacterial chromosome. Adapted from ref. [4]. **b** The distribution and directionality of *attB* sites in *S.* Typhimurium LT2 indicates the potential for transfer of up to 3.5 Mb of chromosomal DNA from a poly-lysogenic background, representing approximately 72% of the bacterial chromosome. Source data are provided as a Source Data file.

non-inducing conditions and providing a mechanism, via recombination in the recipient cell, for the extensive mosaic-like variation observed in the cargo genes of different SaPIs[27].

Though the picture is less striking in the case of *S.* Typhimurium, patterns of virulence-associated genes being located close enough to phage integration sites in the direction of packaging to facilitate their transfer via LT can nevertheless be observed (Table 4). In strain LT2, the 39.7 kb *Salmonella* Pathogenicity Island (SPI)-2, which encodes the genes necessary for intra-macrophage survival and proliferation[28], is located 363.5 and 111.6 kb downstream of prophage attachment sites St3 and St4, respectively, permitting packaging into particles approximately 9–10 (St3) and 4–5 (St4) headfuls from the phage packaging initiation site. SPI-4, encoding the *siiE* non-fimbrial adhesion factor and the *siiABCDF* genes encoding the T1SS responsible for the transfer of SiiE into the host cell[29], is located approximately 6–7 and 3 HFs downstream of the St9 and St10 *attB* sites, respectively. Similarly, SPI-5, which encodes effector proteins that contribute to host cell cytoskeletal rearrangement during invasion and intramacrophage survival[30], is also orientated downstream of prophage attachment sites St2 and St3, permitting packaging into LT particle HF6 and HF3, respectively. Interestingly, the host cell invasion-associated locus, SPI-1, is positioned in the LT2 chromosome such that it should be compatible with LT originating from a prophage integrated at St8 *attB*, though admittedly, as it would be expected to be packaged into LT particle headful 9 or 10, its transfer efficiency may be reduced compared to that of other loci via the LT mechanism, suggesting that LT may be only one of the multiple mechanisms involved in mediating the HGT of these genetic elements.

**A role for LT in maintaining the core genome.** Although LT provides a mechanism for the potential transfer of massive

quantities of bacterial chromosomal DNA at high frequencies, this transfer is likely only of benefit to closely related strains as (i) phages are typically considered to have narrow host-ranges, though the silent transfer of DNA by phages and PICIs has been reported between different species and genera[31–34], suggesting that host-restriction for many phages is not as stringent as previously thought; and, vitally, (ii) donor and recipient cells must share sufficient DNA sequence homology to support homologous recombination (HR) of the transduced DNA into the chromosome. Such limits pose interesting questions as to the overall function of such widespread gene exchange between closely related bacteria and how it impacts evolution at the species level. While it is certainly possible, and indeed plausible, that LT facilitates the exchange of novel virulence genes between related strains, it may be that under most circumstances it serves a less conspicuous but equally important function: purifying the chromosome of undesirable mutations. The true extent of HR in bacterial lineages is difficult to infer as it is technically challenging to fully assess the frequency of such events between closely related organisms where the vast majority of recombination events would be almost impossible to detect, however evidence from MLST and biosynthetic gene sequencing analyses indicate that bacterial species do engage in frequent HR events[35], though this may be to a greater or lesser extent depending on the species[36]. Despite the evidence that HR occurs regularly in nature, fundamental questions remain unsolved regarding both the mechanism of transfer involved in providing exogenous DNA, as well as the volume of this external DNA that arrives at the recipient cells, particularly in species that have not been demonstrated to be naturally transformable. Given the propensity for poly-lysogeny in many bacteria, coupled with the frequency and magnitude of DNA transfer by LT, we propose here that LT facilitates the continual exchange of core genes between strains. This constant arrival of exogenous

**Table 4 Examples of virulence-associated loci compatible with transfer via LT.**

| S. aureus Locus | Contribution to virulence[b] | Compatibility with prophage attachment (attB) sites for packaging by LT[a] | | | | | | | | | |
|---|---|---|---|---|---|---|---|---|---|---|---|
| | | Sa1 | Sa2 | Sa3 | Sa4 | Sa5 | Sa6 | Sa7 | Sa8 | Sa9 | Sa12 |
| vSaα | Lpl (induction of host inflammatory response) | | | | | | HF 3 | | | | |
| vSaβ | LukD, LukE, hysA (toxins) | | | HF 5-6 | | HF 2-3 | | | HF 1-2 | | |
| vSaγ | Hla (toxin) | HF 7 | | | HF 4-5 | | | HF 2 | | | |
| clfA | ClfA (adhesin) | | | | | | | | | HF 1 | HF 6 |
| map | MHC class II analogue protein (immune evasion) | | | HF 1 | | | | | | | |
| SaPI Type I attB | SaPIpT1028, SaPI4 | | | | | | HF 2 | | | | |
| SaPI Type II attB | SaPIbov1 (TSST-1, Sel, Sek); SaPIbov2 (Bap); SaPIbov5 (Scin, vWbp) | | | | | | HF 3 | | | | |
| SaPI Type III attB | SaPIm4 (FhuD); SaPImw2 (Ear, Seb, Sel, Sek); SaPI5 (Ear, Sek, Seq) | | | | | | | | | HF 1 | HF 6 |
| SaPI Type IV attB | SaPI1 (TSST-1, Ear, Sek, Seq); SaPI3 (Ear, Seb, Sel, Sek); SaPI5 (Ear, Sek, Seq) | | | | | | | | | HF 2 | HF 6 |

| S. Typhimurium Locus | Contribution to virulence | Compatibility with prophage attachment (attB) sites for packaging by LT[a] | | | | | | | | | |
|---|---|---|---|---|---|---|---|---|---|---|---|
| | | St1 | St2 | St3 | St4 | St5 | St6 | St7 | St8 | St9 | St10 |
| SPI1 | Invasion of host cells | | | | | | | | HF 9-10 | | |
| SPI2 | Intracellular survival in macrophages | | | HF 9-10 | HF 4-5 | | | | | | |
| SPI4 | SiiABCD and SiiE (adhesin) | | | | | | | | | HF 6-7 | HF 3 |
| SPI5 | SopB (host cell cytoskeletal rearrangement), PipB (intramacrophage survival) | | HF 6 | HF 3 | | | | | | | |
| SPI12 | Intracellular survival in macrophages; systemic survival in mice | | | | | | HF 12 | | | | |
| SPI16 | Contains three putative ORFs with a putative role in O-antigen glycosylation | HF 7 | | | | | | | | | |
| CS54 | Putative island postulated to have a role in adhesion | | | | | | HF 4-5 | HF 7 | | | |
| tnpA_1 | IS200 transposon | | HF 2-3 | | | | | | | | |
| tnpA_2 | IS200 transposon | | | | | HF 3 | | | | | |
| tnpA_3 | IS200 transposon | | | | | | | | HF 8 | | |
| tnpA_4 | IS200 transposon | | | | | | | | | HF 6 | |
| tnpA_6 | IS200 transposon | | | | | | | | | HF8 | HF5 |

[a]HF denotes the expected LT particle headful that the element would be expected to be packaged into by a standard-sized phage (~45 kb for staphylococcal phages, ~43 kb for *Salmonella* phages).
[b]*lpl* lipoprotein-like, *lukD* leukotoxin D, *lukE* leukotoxin E, *hysA* hyaluronate lyase, *hla* α–haemolysin, *tst* toxic shock syndrome toxin, *sel* staphylococcal enterotoxin L, *sek* staphylococcal enterotoxin K, *bap* biofilm-associated protein, *scin* staphylococcal complement inhibitor, *vWbp* von Willebrand binding protein, *ear E. coli* ampicillin resistance, *seq* staphylococcal enterotoxin Q, *sel* staphylococcal enterotoxin L, *seb* staphylococcal enterotoxin B.
Source data are provided as a Source Data file.

DNA can be used to purify the chromosome of undesirable mutations, reducing deleterious genetic drift.

Indeed, there is already evidence in the literature to indicate that incorporation of exogenous DNA via HR can serve as a chromosomal sanitising strategy in some naturally competent species, with Streptococci utilising this mechanism to cure their genomes of parasitic MGEs, including prophages[37]. Such a scenario sets up an interesting dichotomy for the fate of prophages in cells receiving exogenous chromosomal DNA via LT particles, as there is a possibility that resident prophages could be sanitised from the recipient host chromosome if incoming DNA originated from a donor cell that lacked a prophage at the corresponding chromosomal attachment site. It is likely in this scenario that the role of the prophage in contributing to host niche adaptation has an important part to play in its retention; prophages in *Streptococcus pneumoniae* typically do not carry accessory genes of apparent benefit to the bacterial host cell[37] rendering prophages in this species overt parasites. In contrast, many staphylococcal prophages contribute genes with important virulence traits to their *S. aureus* hosts[38], tipping the selective balance in favour of prophage maintenance by the host cell. Furthermore, it is important to note that despite the presence of competence genes in more than 80 bacterial species[39], relatively few bacterial species have been demonstrated to actively engage in natural transformation, so it may be that natural transformation and LT serve parallel functions in different bacterial species depending on their tendency towards or against lysogeny. To date, LT has been described only for phages of *S. aureus* (80α and φ11[16]), *S.* Typhimurium (P22) and *Enterococcus faecalis* (φp1)[17], and thus it is likely that as more bacterial-phage pairs are studied, we will glean more information on the implications of LT-mediated chromosomal curing for prophage distribution in other species.

In addition to providing substrate DNA for curing disadvantageous mutations of the chromosome, it is also possible that LT, with subsequent HR of the transduced DNA, has an additional important role to play in mitigating the effect of clonal interference in polymicrobial communities. HR of exogenous DNA has previously been shown to be important for escaping clonal interference in drug-resistant bacterial populations, whereby HR enables multiple independent advantageous mutations to be transferred from different competing donor organisms and combined into a single recipient, bestowing upon it a competitive advantage over the donor populations carrying the

**Table 5 Bacterial strains and plasmids.**

| Strain | Description | Reference |
|---|---|---|
| RN4220 | *Staphylococcus aureus* restriction-defective derivative of RN450 | Lab strain |
| JP6399 | RN450 lysogenic for 80α carrying an erythromycin resistance cassette (80α::*ermC*) | 16,57 |
| JP6400 | RN451 lysogenic for φ11 carrying an erythromycin resistance cassette (φ11::*ermC*) | 16,57 |
| JP14277 | RN4220 SAOUHSC_01121::*cad*CA; cadmium resistance cassette inserted 35 kb downstream of Sa7 *att*B | 16 |
| JP19145 | RN4220 SAOUHSC_01121::*cad*CA; cadmium resistance cassette inserted 5 kb downstream Sa5 *att*B | 16 |
| JP20714 | JP14277 pJP2511 | This work |
| JP20716 | JP19145 pJP2511 | This work |
| JP20844 | JP20714 lysogenic for φ11::*ermC* | This work |
| JP20846 | JP20716 lysogenic for 80α::*ermC* | This work |
| JP18938 | *Salmonella enterica* serovar Typhimurium LT2 ΔFels-1 ΔGifsy-2 ΔGifsy-1 ΔFels-2 | 17 |
| JP18983 | JP18938 lysogenic for P22 | 17 |
| JP22210 | JP18983 lysogenic for P22 *sie::cat* | This work |
| Plasmid | Description | Reference |
| pKD46 | Thermosensitive plasmid with Red lambda system, *amp*R | 58 |
| pJP2511 | Gram-positive plasmid containing a chloramphenicol resistance cassette, *cat*R | 59 |

individual mutations[40]. Given the capacity for large spans of DNA to be transferred at high frequency via LT, it is likely that LT also contributes to genome evolution via this mechanism.

**Gene transfer agents: pretenders to the LT throne?** In some species, including *Rhodobacter capsulatus*, *Bartonella* spp. and *Bacillus* spp., horizontal transfer of chromosomal DNA may also be facilitated by small phage-like particles knowns as gene transfer agents (GTAs)[41]. The exact role of GTAs remains unclear, though these elements do share some similar characteristics with transducing particles, principally in that they package and transfer chromosomal DNA from their donor cell in a tailed, membrane-bound vesicle reminiscent of phage virions, with the tail on the GTA particle likely demanding receptor specificity for the recipient host cell, and they do not readily transfer genes required for their own propagation[41]. However, GTAs and LT particles also exhibit stark dissimilarities, with implications for their respective putative roles in contributing to bacterial genome evolution via HGT. While GTAs package chromosomal DNA from their donor cell, they do so in an apparently random manner[41], which also occurs in GT but is in contrast to the specificity in packaging observed in LT, directed by the prophage integration site. In addition, in situ replication of activated prophages during the early LT process leads to amplification of adjacent chromosomal DNA, resulting in the generation of multiple copies of the chromosomal genes downstream of the prophage[16]. Such replication ensures that there is an excess of substrate DNA for packaging via the LT mechanism, generating a large pool of LT particles containing similar regions of DNA sequence that may then be transferred to new recipient cells at high frequency, optimising the chances of successful HR occurring. By contrast, to our knowledge, there is currently no available data to suggest that such redundancy in the packaged DNA occurs for GTAs. In addition, GTAs also appear to be limited in terms of their packaging capacity, transferring only ~5–15 kb of DNA per virion particle[41], which is significantly lower than the capacity available for transfer via LT particles, which exhibit capacities comparable to that of their facilitating prophage. Taken together, owing to the differences in the magnitude and specificity of DNA transferred by GTAs and LT, it is arguably unlikely that GTAs represent a mechanism for the HGT of DNA with equivalent power to that of LT.

## Concluding remarks

In summary, we propose here that the discovery of LT has redefined how we perceive genetic mobility in bacteria. This powerful mechanism of transduction deconstructs the conventional sense of an MGE by uncoupling mobility from the genetic element and imparting it to the bacterial genome, where mobility is designated by coordinates in the chromosome (relative to prophages) and not defined by the carriage of the MGEs themselves. In effect, LT creates highly efficient platforms for chromosomal packaging and transfer that facilitates the exchange of 'immobile' core genes at high frequencies that are significantly higher than those of most elements classically considered as mobile. Furthermore, such dramatic exchange of chromosomal DNA not only provides significant opportunities for the rapid acquisition of virulence factors but also offers a plausible mechanism for the delivery of core genetic material between related strains at sufficiently high rates to permit the extensive HR necessary to maintain the species identity. Surprisingly, our analysis indicated that many MGEs are not as mobile as we previously assumed and that despite the existence of strategies to enhance their transmissibility, their transfer is restricted. We recognise that what we propose here is heretical to the accepted paradigm of bacterial genetic mobility, and we would be remiss to ignore the argument that because chromosomal mobility is completely dependent on other members of the mobilome, namely phages, it cannot truly be considered mobile. For that, we would contend that this standpoint is undermined by the broad inclusion of non-conjugative plasmids and transposons as members of the mobilome. Ultimately, the discovery of LT has blurred the lines of what we can consider being mobile, and therefore there is a need for a scientific discussion about whether the bacterial chromosome itself could actually qualify as a mobile element.

## Methods

**Bacterial strains and plasmids**. The bacterial strains and plasmids used in this study are detailed in Table 5. *S. aureus* strains were grown in tryptic soy broth (TSB) or on tryptic soy agar (TSA) plates supplemented with 1.7 mM sodium citrate. *Salmonella enterica* serovar Typhimurium was grown in Luria Bertani (LB) broth or on LB agar (LBA) plates. Where appropriate, the following antibiotic concentrations were used for phage, plasmid, or chromosomal marker selection: erythromycin, 10 μg/ml; ampicillin, 100 μg/ml; chloramphenicol, 20 μg/ml; cadmium chloride, 100 μM.

**Construction of marked phage P22**. Insertion of a chloramphenicol resistance marker into *Salmonella* phage P22 was performed as previously described[7]. Briefly, polymerase chain reaction (PCR) amplification of the chloramphenicol resistance marker was performed using oligonucleotides P22-sie-cat-1m (5′-GGCAAAGTAC CACACTGTTATCAGCAAACTTAGTGGTATATGAATTACTGGGAATAGGAA CTTCATTTAAATGGC) and P22-sie-cat-2c (5′-GCGCACTCATGGGCAATAGTA GATAGTTTGCCATTGAACACGCCTATCACGGGCGCGCCTACCTGTGACGG). PCR products were transformed into recipient strain JP18983 harbouring plasmid

pKD46, which expresses the λ Red recombinase, facilitating the insertion of the marker into the phage genome. The placement of the marker was subsequently verified by PCR and Sanger sequencing (Eurofins Genomics).

**Phage induction**. *S. aureus* strains lysogenic for phages φ11::*ermC* and 80α::*ermC*, were grown in TSB to early exponential phase ($OD_{540}$ ~0.15) at 37 °C and 120 rpm. *S.* Typhimurium lysogenic for phage P22*sie::cat*, was grown in LB to early exponential phase ($OD_{600}$ ~0.20) at 37 °C and 120 rpm. All prophages were induced by the addition of mitomycin C (2 µg/ml), then were incubated for 4–5 h at 30 °C followed by overnight incubation at room temperature to permit full phage lysis, before filtering with a 0.2 µm syringe filter (Sartorius).

**Determination of phage lysogenisation rates**. To determine the rate of lysogenisation for *S. aureus* phages 80α and φ11, 100 µl of phage lysate or a dilution of lysate prepared in phage buffer (100 mM NaCl, 50 mM Tris, 10 mM $MgSO_4$, 4 mM $CaCl_2$) was used to infect 1 ml of recipient *S. aureus* RN4220 cells at $OD_{540}$ 1.4, supplemented with 4.4 mM $CaCl_2$. After static incubation for 20 min at 37 °C, 3 ml TTA (TSA top agar; 3 g/100 ml TSB, Oxoid, plus 0.75 g/100 ml agar, Formedium) was added to each reaction and the entire contents were poured onto a TSA plate supplemented with 1.7 mM sodium citrate +10 µg/ml erythromycin.

The same method was used for determination of the transfer rate of plasmid pJP2511 and $Cd^R$ chromosomal markers into RN4220 by phage 80α and φ11 transduction, with selection on TSA supplemented with 1.7 mM sodium citrate + 20 µg/ml chloramphenicol (for the plasmid) or 100 µM $CdCl_2$ (for chromosomal markers).

To determine the rate of lysogenisation for *S.* Typhimurium phage P22, 100 µl of phage lysate or a dilution of lysate prepared in phage buffer was used to infect 1 ml of recipient *Salmonella* (JP18938) cells at $OD_{600}$ 1.4, supplemented with 4.4 mM $CaCl_2$. After static incubation for 20 min at 37 °C, 3 ml LTA (LB top agar; 2 g/100 ml LB, Oxoid, plus 0.75 g/100 ml agar, Formedium) was added to each reaction and the entire contents were poured onto an LBA plate supplemented with 20 µg/ml chloramphenicol.

In all cases, plates were incubated at 37 °C for 24 h and the number of colonies formed (representing phage particles present in the lysate) was counted and expressed as the colony-forming units (CFU)/ml. Results are reported as the number of lysogens or transductants obtained per donor cell-induced (for *S. aureus* this is equivalent to $6.5 \times 10^7$ donor CFU/ml at $OD_{540}$ 0.15; for *Salmonella* this is equivalent to $1 \times 10^8$ donor CFU/ml at $OD_{600}$ 0.20) and are the mean values of three (P22) or four (80α and φ11) independent experiments.

**Analysis of transfer rates**. Transfer frequency values for the different elements analysed were obtained from the published literature, with their sources indicated in Tables 1 and 3. In order to enable comparisons between conjugation and the other modes of horizontal gene transfer, the transfer frequency of published phage- and PICI-mediated DNA transfer was re-analysed as TE per bacterial donor cell at the time of prophage induction: TE per donor cell = Lysogenisation or Transductant Units (TrU) per ml/donor CFU per ml at the time of phage induction. For the *S. aureus* data analysed, lysogens were reportedly induced at $OD_{540}$ ~0.15, which is equivalent to $6.5 \times 10^7$ CFU per ml in the donor population, while for *Salmonella*, lysogens were induced at $OD_{600}$ 0.2, which is equivalent to $1 \times 10^8$ CFU per ml in the donor population.

**Estimation of the cargo capacity**. SnapGene Viewer version 5.3 (open source software: https://www.snapgene.com/snapgene-viewer/) was used to import and visualise DNA sequences from Genbank for different genetic elements where accession numbers were available, to enable determination of the total number of ORFs and their predicted functions for calculation of estimated 'cargo capacity' rates. The accession numbers are provided in the manuscript in Tables 1 and 3. This programme was also used to import and visualise complete chromosomal DNA sequences for *S. aureus* NCTC8325-4 (accession: NC_007795.1) and *S.* Typhimurium LT2 (accession: AE006468.2) to enable: (i) mapping of the coordinates of successive lateral transducing particle headfuls from each *attB* site; (ii) estimation of the number of ORFs contained within each lateral transducing particle headful for calculation of estimated 'cargo capacity'; (iii) location mapping of loci/genes of interest relative to *attB* sites on the *S. aureus* and *S.* Typhimurium chromosomes.

**Reporting summary**. Further information on research design is available in the Nature Research Reporting Summary linked to this article.

## Data availability

Source data is available in the accompanying excel file. For all data derived from published literature, the source is indicated as a citation in the manuscript. Accession numbers relating to each element analysed are included in the manuscript (Tables 1 and 3), where they were available. Source data are provided with this paper.

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

## Acknowledgements

We thank Jean-Marc Ghigo, Álvaro San-Millán, David W. Holden and Julian Parkhill for comments on this paper. This work was supported by grants MR/M003876/1, MR/V000772/1 and MR/S00940X/1 from the Medical Research Council (UK), BB/N002873/1, BB/V002376/1 and BB/S003835/1 from the Biotechnology and Biological Sciences Research Council (BBSRC, UK), ERC-ADG-2014 Proposal n° 670932 Dut-signal (from EU), to J.R.P; and Wellcome Trust 201531/Z/16/Z to J.R.P.

## Author contributions

S.H. and J.R.P. conceived the study. S.H. performed the data analysis. A.F.S., N.Q.P., R.I.C. and A.H. performed the experiments. S.H., J.C. and J.R.P. wrote the paper. J.R.P. supervised the work.

## Competing interests

The authors declare no competing interests.
