## [Peer Review File · Nature Communications]

Bacterial chromosomal mobility via lateral transduction exceeds that of classical mobile genetic elementsREVIEWER COMMENTS

Reviewer #1 (Remarks to the Author):

In this manuscript by Humphrey et al., the authors aggregate data from several published sources as well as new experimental work to compare different mechanisms of gene exchange in both *S. aureus* and *S. Typhi*. The authors put forward the argument that lateral transduction (whereby prophage-adjacent chromosomal DNA is sequentially-packaged into phage capsids) is a more efficient means of within-species horizontal gene transfer than conjugation or generalised transduction, implying that chromosomal loci may be more susceptible to mobilisation than classical 'mobile genetic elements'. Though I don't fully agree with the authors on every point, I think our disagreements represent a matter of perspective rather than scientific fact, and I think this work makes a valuable and provocative contribution to the literature, with broad interest to researchers in microbiology and microbial evolution. However, I do have several suggestions, as well as criticisms which should be addressed in future revisions:

1. I found the tabular presentation of the data to be difficult to assess easily. It is hard to compare between different HGT mechanisms. It would be much clearer were the data presented as plots (with log-transformed y-axis), and the table provided as supplementary information.
2. The concluding argument is that the bacterial chromosome itself could qualify as a mobile element, owing to its propensity to transfer by HGT. This misrepresents the key feature of mobile genetic elements: they have distinct (selfish) fitness interests because they have histories (and fates) that are distinct from other loci. The same is not true for the chromosome as a whole. However, it may be true for certain chromosomal loci. In fact, the authors' argument — that prophage-adjacent genes are subject to higher rates of mobilisation than even conjugative plasmids — generates testable hypotheses along these lines, namely, that these prophage-adjacent regions should resemble the gene content of mobile genetic elements. For example, one might expect the transferred regions to become enriched in certain kinds of genes relative to the rest of the chromosome, such as locally-adaptive accessory genes (*sensu* Eberhard 1990 doi: 10.1016/0147-619x(89)90040-1), and/or parasitic DNA such as transposons. Are they? (Indeed, could these preferentially laterally-transduced regions provide some origin for PICIs — regions that were originally transduced passively have now evolved to exploit phage transmission completely?) Chromosomal regions susceptible to lateral transduction should also show footprints of recombination (i.e. less linkage). An analysis of genome evolution in light of the (purported) predominance of lateral transduction could provide orthologous evidence for the authors' argument, greatly enhancing the impact of this study. The genome sequences are available for such analyses, but it would represent considerable work. At the very least the authors should speculate on the implications of their findings for genome evolution.
3. The authors should briefly discuss gene transfer agents (GTAs), which have some similar and some different effects to laterally-transducing particles. See Lang et al. 2012 (doi: 10.1038/nrmicro2802).
4. Homologous recombination in Streptococci does indeed appear to 'sanitise' the chromosomes of undesired mutations, namely prophage (see Croucher et al. 2016 doi: 10.1371/journal.pbio.1002394) — this would set up an interesting conflict when considering that it may be laterally-transducing prophage that enable this sanitation in the first places (Sa8, Sa4, St4 seem particularly at risk)! This could be discussed.
5. Besides 'santitising the chromosome', I imagine that LT could also play an important role in escaping from clonal interference, whereby multiple beneficial mutations compete. See e.g. Perron et al. 2011 doi: 10.1098/rspb.2011.1933.

Minor suggestions:

1. On the whole, the current work aggregates previously published data, with only a small minority of the transfer rates that are presented representing new data. The manuscript may therefore be better suited to a more explicitly Perspective/Opinion/Review-type of journal.
2. First paragraph of Introduction is lacking references.
3. Last paragraph of the Results section, the authors imply that "preserving species identity" is as important to bacteria as acquiring new adaptive genes. Surely 'species identity' is important only to (human) taxonomists; I don't imagine that bacteria care?!

4. "HR has been proposed as a mechanism for ensuring taxonomic consistency⁶" — check reference 6, I'm not sure it supports this statement.

Reviewer #2 (Remarks to the Author):

In this manuscript, Humphrey et al. propose that bacterial genomes should be considered as MGEs in light of the recent discovery of lateral transduction, whereby large regions of chromosomal DNA can be packaged into phage capsids and disseminated. Through a series of comparative experiments and previously published literature searches, the authors show that laterally transduced DNA can transfer at efficiencies equal to or greater than well studied MGEs such as conjugative plasmids, ICEs, etc. Using *S. aureus* and *S. enterica* as models, they suggest that lateral transduction through the identification of potential att sites within the chromosome could transfer large portions of bacterial genomes, upwards of Mb amounts across multiple headfuls of phages. The authors argue that this discovery should refocus the discussion of what it means to be an MGE and to extend this definition beyond plasmids, phages, and ICEs to also include the bacterial chromosome as mobile via lateral transduction.

I have the following comments for the authors moving forward.

1. The general tone of this manuscript reads as if lateral transduction applies broadly to most bacteria-phage pairs. Although, this might be and is likely to be the case, for now it is my understanding that this has only been tested in Staph and Salmonella. I suggest making this point very clear to the reader.
2. It is interesting that lateral transduction is more efficient for Staph than Salmonella. Can the authors comment on why this might be?
3. Are the polylysogeny sites shown in Fig. 2 proven, or are they only predicted based on att site sequence homology. If the latter, verifying a few of these as legitimate insertion sites that can prime lateral transduction events would greatly strengthen their argument about the breadth of DNA transfer inferred in this study.
4. On lines 198 – 199 the authors state that sufficient DNA homology in the receiving host would be needed to make these lateral transduction events productive. However, have the authors considered homology facilitated illegitimate recombination, whereby only micro homology would be needed for crossover events, albeit at lower frequencies?

Reviewer #3 (Remarks to the Author):

The work builds upon the previous report of lateral transduction by Staphylococcus phages. Here the authors have used some very eloquent work to compare the frequency of HGT via different mechanisms (conjugation, SaPI, specific transduction, GT) and compared this to the rates by lateral transduction. They have combined the previously published rates with derived frequencies in this study. This is the first time the frequency of HGT from different mechanisms has been compared in such a way. By doing this for both Gram -ve and Gram +ve bacteria they have firstly demonstrated lateral gene transfer is not just limited to Staph. Of more significance they have comprehensively compared this to the rates of other mechanisms of HGT. Demonstrating that LGT exceeds the frequency of other HGT mechanisms in most cases and the proportion of the genome that can be mobilized by LGT is far greater than other mechanisms. This important finding, as highlighted by the authors raises the question of exactly what proportion of the genome is mobile. It provides clear evidence that far more of the genome is more mobile than previously thought. The identification that such high proportions of the genome are mobile, is an important finding, with a wide impact. The authors go on to speculate the large scale HGT from LGT may be involved

in maintenance of bacterial species. This opens up debate on the role of HGT and what should be considered mobile. The results will change the way phages are perceived to have a role in HGT, with phage mediated HGT often thought to be of less importance than other mechanisms. It will likely result in many further studies in other systems that look at the frequency of LGT from prophages. The methodology is well carried out and clearly described, allowing for the work to be repeated. The paper is well written and a pleasure to read.

Reviewer #1 (Remarks to the Author):

In this manuscript by Humphrey et al., the authors aggregate data from several published sources as well as new experimental work to compare different mechanisms of gene exchange in both *S. aureus* and *S. Typhi*. The authors put forward the argument that lateral transduction (whereby prophage-adjacent chromosomal DNA is sequentially-packaged into phage capsids) is a more efficient means of within-species horizontal gene transfer than conjugation or generalised transduction, implying that chromosomal loci may be more susceptible to mobilisation than classical 'mobile genetic elements'. Though I don't fully agree with the authors on every point, I think our disagreements represent a matter of perspective rather than scientific fact, and I think this work makes a valuable and provocative contribution to the literature, with broad interest to researchers in microbiology and microbial evolution. However, I do have several suggestions, as well as criticisms which should be addressed in future revisions:

1. I found the tabular presentation of the data to be difficult to assess easily. It is hard to compare between different HGT mechanisms. It would be much clearer were the data presented as plots (with log-transformed y-axis), and the table provided as supplementary information.

We agree that the representation of all the data is difficult, mainly because the number of elements analysed as well as the huge variation observed in the transfer of these elements. After trying many different possibilities, we arrived at the conclusion that presenting the data as Tables, which also contain additional but important information, is the best way to provide these results.

2. The concluding argument is that the bacterial chromosome itself could qualify as a mobile element, owing to its propensity to transfer by HGT. This misrepresents the key feature of mobile genetic elements: they have distinct (selfish) fitness interests because they have histories (and fates) that are distinct from other loci. The same is not true for the chromosome as a whole. However, it may be true for certain chromosomal loci. In fact, the authors' argument — that prophage-adjacent genes are subject to higher rates of mobilisation than even conjugative plasmids — generates testable hypotheses along these lines, namely, that these prophage-adjacent regions should resemble the gene content of mobile genetic elements. For example, one might expect the transferred regions to become enriched in certain kinds of genes relative to the rest of the chromosome, such as locally-adaptive accessory genes (*sensu* Eberhard 1990 doi: 10.1016/0147-619x(89)90040-1), and/or parasitic DNA such as transposons. Are they? (Indeed, could these preferentially laterally-transduced regions provide some origin for PICs — regions that were originally transduced passively have now evolved to exploit phage transmission completely?) Chromosomal regions susceptible to lateral transduction should also show footprints of recombination (i.e. less linkage). An analysis of genome evolution in light of the (purported) predominance of lateral transduction could provide orthologous evidence for the authors' argument, greatly enhancing the impact of this study. The genome sequences are available for such analyses, but it would represent considerable work. At the very least the authors should speculate on the implications of their findings for genome evolution.

This was a very nice comment, and we can confirm that the reviewer was right. Therefore, many chromosomally-encoded pathogenicity islands localise in the chromosome so they can be mobilised by LT. We thank the reviewer for their suggestion. We have added a section in the *Results and Discussion* (lines 188-264) addressing the reviewer's query.

3. The authors should briefly discuss gene transfer agents (GTAs), which have some similar and some different effects to laterally-transducing particles. See Lang et al. 2012 (doi: 10.1038/nrmicro2802).

We have added a section (lines 324-350) to discuss the role of GTAs in HGT and their similarities and differences to LT particles.

4. Homologous recombination in Streptococci does indeed appear to 'sanitise' the chromosomes of undesired mutations, namely prophage (see Croucher et al. 2016 doi: 10.1371/journal.pbio.1002394) — this would set up an interesting conflict when considering that it may be laterally-transducing prophage that enable this sanitation in the first places (Sa8, Sa4, St4 seem particularly at risk)! This could be discussed.

We thank the reviewer for their insight with this suggestion. We have discussed this idea in a new section (lines 265-313) in the *Results and Discussion*.

5. Besides 'santitising the chromosome', I imagine that LT could also play an important role in escaping from clonal interference, whereby multiple beneficial mutations compete. See e.g. Perron et al. 2011 doi: 10.1098/rspb.2011.1933.

Again, we thank the reviewer for their insight. We have added a paragraph (lines 314-323) to address this point.

Minor suggestions:

1. On the whole, the current work aggregates previously published data, with only a small minority of the transfer rates that are presented representing new data. The manuscript may therefore be better suited to a more explicitly Perspective/Opinion/Review-type of journal.

The manuscript was originally intended as a Perspective/Opinion article, however it was suggested to us that we should consider submission to *Nature Communications* as a full article due to the nature of the analysis involved.

2. First paragraph of Introduction is lacking references.

We thank the reviewer for pointing this out. These references were originally omitted due to severe constraints on the number of references permitted for Perspective type articles. We have now added the appropriate references back in.

3. Last paragraph of the Results section, the authors imply that "preserving species identity" is as important to bacteria as acquiring new adaptive genes. Surely 'species identity' is important only to (human) taxonomists; I don't imagine that bacteria care?!

This is a very interesting point for discussion. One can argue that bacteria have evolved mechanism that will keep the species identity (i.e. LT). One can also argue that with lower rates of transfer than those observed by LT, bacteria could transfer relevant genes, which in the appropriate recipient strains will expand by clonal expansion of the recipient cells. This is what occurs with the classical MGEs. Therefore, we are quite convinced that the major role of LT is purifying the core genome of undesirable mutations, reducing deleterious genetic drift (or in other words, preserving species identity). In any case, we have now slightly changed this paragraph, to make our rationale a bit clearer.

4. "HR has been proposed as a mechanism for ensuring taxonomic consistency" — check reference 6, I'm not sure it supports this statement.

We thank the reviewer for their attention to detail in noticing this error. We have re-written the affected section (lines 277-283) in order to address this.

Reviewer #2 (Remarks to the Author):

In this manuscript, Humphrey et al. propose that bacterial genomes should be considered as MGEs in light of the recent discovery of lateral transduction, whereby large regions of chromosomal DNA can be packaged into phage capsids and disseminated. Through a series of comparative experiments and previously published literature searches, the authors show that laterally transduced DNA can transfer at efficiencies equal to or greater than well studied MGEs such as conjugative plasmids, ICEs, etc. Using *S. aureus* and *S. enterica* as models, they suggest that lateral transduction through the identification of potential att sites within the chromosome could transfer large portions of bacterial genomes, upwards of Mb amounts across multiple headfuls of phages. The authors argue that this discovery should refocus the discussion of what it means to be an MGE and to extend this definition beyond plasmids, phages, and ICEs to also include the bacterial chromosome as mobile via lateral transduction.

I have the following comments for the authors moving forward.

1. The general tone of this manuscript reads as if lateral transduction applies broadly to most bacteria-phage pairs. Although, this might be and is likely to be the case, for now it is my understanding that this has only been tested in Staph and Salmonella. I suggest making this point very clear to the reader.

The reviewer is correct that to date, LT has been only described in the published literature for *S. aureus* phages. We currently have a paper in revision that indicates that both *Salmonella* Typhimurium phage P22 and *Enterococcus faecalis* phage ϕ p1 also engage in LT. We have clarified this point in the manuscript – lines 309-310.

2. It is interesting that lateral transduction is more efficient for Staph than Salmonella. Can the authors comment on why this might be?

We believe that this is an effect of the difference in the efficiency of *Salmonella* phage P22 replication during the infectious cycle vs following prophage induction. From the raw data, we obtained phage titre values for *Salmonella* phage P22 that were ~1.5 logs higher following phage infection (2.05×10^{10} PFU/ml \pm 2.63×10^9) than were obtained following phage induction (6.6×10^8 PFU/ml \pm 5.6×10^7). By contrast, the phage titres obtained for *S. aureus* phage 80 α is approximately equal for both induction (1.42×10^{10} PFU/ml \pm 7.46×10^9) and infection (1.31×10^{10} PFU/ml \pm 7.46×10^9) so the difference between the two transduction mechanisms (induction for LT, infection for GT) is easier to compare. These data may suggest that P22 is either not as efficient as phage 80 α at being induced while in the prophage state, or that it does not engage in *in situ* replication and/or packaging as efficiently as 80 α , though further studies would be required to elucidate the exact mechanism at play to explain such differences.

3. Are the polylysogeny sites shown in Fig. 2 proven, or are they only predicted based on att site sequence homology. If the latter, verifying a few of these as legitimate insertion sites that can prime lateral transduction events would greatly strengthen their argument about the breadth of DNA transfer inferred in this study.

In this figure we indicated all the functional *attB* sites present in either *S. aureus* or *Salmonella* Typhimurium. It does not mean that a single strain has prophages integrated in all the *attB* sites. But definitely, these sites are conserved and functional. While in the *Salmonella* study the two phages used (P22 and ES18) integrate in the same *attB* site, in the study describing LT in *S. aureus* we already used prophages that integrate at different *attB* sites, confirming that large parts of the bacterial chromosome can be mobilised at high frequencies by LT.

4. On lines 198 – 199 the authors state that sufficient DNA homology in the receiving host would be needed to make these lateral transduction events productive. However, have the authors considered homology facilitated illegitimate recombination, whereby only micro homology would be needed for crossover events, albeit at lower frequencies?

This is a very interesting question that we have recently addressed in the lab. We think that this illegitimate recombination can be used by phages to create variation (note that the first particles created by LT contain half of the phage genome). We also want to explore whether chromosomal genes could be mobilised among different species by LT. Unfortunately, we do not have data yet supporting any of these ideas.

Reviewer #3 (Remarks to the Author):

The work builds upon the previous report of lateral transduction by Staphylococcus phages. Here the authors have used some very eloquent work to compare the frequency of HGT via different mechanisms (conjugation, SaPI, specific transduction, GT) and compared this to the rates by lateral transduction. They have combined the previously published rates with derived frequencies in this study. This is the first time the frequency of HGT from different mechanisms has been compared in such a way. By doing this for both Gram –ve and Gram +ve bacteria they have firstly demonstrated lateral gene transfer is not just limited to Staph. Of more significance they have comprehensively compared this to the rates of other mechanisms of HGT. Demonstrating that LGT exceeds the frequency of other HGT mechanisms in most cases and the proportion of the genome that can be mobilized by LGT is far greater than other mechanisms. This important finding, as highlighted by the authors raises the question of exactly what proportion of the genome is mobile. It provides clear evidence that far more of the genome is more mobile than previously thought. The identification that such high proportions of the genome are mobile, is an important finding, with a wide impact. The authors go on to speculate the large scale HGT from LGT may be involved in maintenance of bacterial species. This opens up debate on the role of HGT and what should be considered mobile. The results will change the way phages are perceived to have a role in HGT, with phage mediated HGT often thought to be of less importance than other mechanisms. It will likely result in many further studies in other systems that look at the frequency of LGT from prophages. The methodology is well carried out and clearly described, allowing for the work to be repeated. The paper is well written and a pleasure to read.

We thank the reviewer for their very kind and generous response.

REVIEWERS' COMMENTS

Reviewer #1 (Remarks to the Author [including 3 attached files]):

The revised version is greatly improved and I think will make a big impact on our understanding of bacterial genome evolution. It is provocative, which is a good thing, provided that readers clearly understand the authors' arguments. For these reasons I make some further suggestions, these are all relatively minor.

Figures & Tables: I'm disappointed that the authors decided to continue with the tabular presentation of their data. This is entirely their decision, but in my opinion a study like this would really benefit from a clear impactful figure(s) demonstrating the efficacy of LT. As it stands, the key information remains hard to parse, particularly as it requires comparison of small superscript-formatted exponents. I would like to give them another opportunity to address this. Regardless of what they decide, I am grateful that the authors provide the information as an Excel spreadsheet, as it allows readers to investigate themselves. In fact I did just that and attach a draft plot of Table 1 as an example that the authors are welcome to use and adapt (I include the relevant R code). I hope the authors do not find this out-of-order — it was done entirely out of interest in and respect for their work — and regardless of their decision I won't push the issue any more.

Lines 191ff. I appreciate the new section on virulence evolution, but it would benefit from some clarification. First, as the authors show, the key fact is that parts of the chromosome are more mobile than others, rather than the chromosome as a whole being highly mobile. Therefore I'd rephrase line 195 as follows: "If we are to assume that parts of the chromosome itself behave akin to classical MGEs..." (line 195). Second, it doesn't follow that genes that enhance intercellular transmission will provide competitive advantage to the cell as a whole (or vice-versa), as is currently implied (lines 197-200). Rather, the authors should explain that MGEs are known to carry distinct genetic cargos that are enriched in genes that provide locally-adaptive traits like toxin production, resistance, and virulence (Eberhard, Plasmid, 1989; DiCenzo & Finan, *Mic Mol Biol Rev*, 2017), relative to the chromosome (as a whole). This is because those genes benefit (selfishly) from being able to move into new genetic backgrounds (see Bergstrom et al. *Genetics* 2000; and especially Niehus et al. *Nat. Comms.* 2015). If parts of the chromosome are indeed hypermobile, theory suggests they should resemble MGEs in terms of gene content. These changes will make the authors' subsequent arguments clearer.

Line 43. Beware of absolutes. Plasmids can also transmit by transduction or by vesicles. Perhaps "Conjugation is the main process by which..." is better.

Lines 44-55 describe conjugative elements (plasmids and ICE). A couple of words might help signpost the reader at the end of this paragraph. For example, describing how conjugative machinery has commonly been held to be the dominant mechanism by which HGT is achieved. See e.g. Halary et al. 2010 (doi: 10.1073/pnas.0908978107)

Line 78. A definition of 'power' is required. In terms of rate of transduction, or material transferred, or...?

Line 85. Successful horizontal gene transfer requires both transmission of DNA, and then maintenance in the recipient. For plasmids, conjugative transposons, and integrative phage, this is achieved through origins of replication and/or integrases (or recombinases/transposases, etc). For LT it is achieved through recipient-encoded recombination processes (homologous recombination, or NHEJ?). A word on this here will be useful to the reader, e.g. "...which are subsequently transferred at high frequencies (Figure 1) and can then be integrated into the recipient genome by homologous recombination." This will help to explain the qualitative differences between LT-mediated chromosomal HGT, and MGE-mediated HGT.

Line 246. Typo ('by facilitated').

Line 276. Is there any evidence that LT-transferred DNA can integrate by NHEJ?

Reviewer #2 (Remarks to the Author):

The authors have adequately addressed my previous inquiries.

Breck A. Duerkop

REVIEWERS' COMMENTS

Reviewer #1:

The revised version is greatly improved and I think will make a big impact on our understanding of bacterial genome evolution. It is provocative, which is a good thing, provided that readers clearly understand the authors' arguments. For these reasons I make some further suggestions, these are all relatively minor.

We would like to thank this reviewer for such nice words, and for their interest in our work.

Figures & Tables: I'm disappointed that the authors decided to continue with the tabular presentation of their data. This is entirely their decision, but in my opinion a study like this would really benefit from a clear impactful figure(s) demonstrating the efficacy of LT. As it stands, the key information remains hard to parse, particularly as it requires comparison of small superscript-formatted exponents. I would like to give them another opportunity to address this. Regardless of what they decide, I am grateful that the authors provide the information as an Excel spreadsheet, as it allows readers to investigate themselves. In fact I did just that and attach a draft plot of Table 1 as an example that the authors are welcome to use and adapt (I include the relevant R code). I hope the authors do not find this out-of-order — it was done entirely out of interest in and respect for their work — and regardless of their decision I won't push the issue any more.

It was really nice that the reviewer spent time to design such nice figure. Following his/her suggestions, we have created now a set of similar figures that clearly show the impact and efficiency of LT (compared to the other mechanisms or processes or gene transfer).

Lines 191ff. I appreciate the new section on virulence evolution, but it would benefit from some clarification. First, as the authors show, the key fact is that parts of the chromosome are more mobile than others, rather than the chromosome as a whole being highly mobile. Therefore I'd rephrase line 195 as follows: "If we are to assume that parts of the chromosome itself behave akin to classical MGEs..." (line 195).

Done.

Second, it doesn't follow that genes that enhance intercellular transmission will provide competitive advantage to the cell as a whole (or vice-versa), as is currently implied (lines 197-200). Rather, the authors should explain that MGEs are known to carry distinct genetic cargos that are enriched in genes that provide locally-adaptive traits like toxin production, resistance, and virulence (Eberhard, Plasmid, 1989; DiCenzo & Finan, Mic Mol Biol Rev, 2017), relative to the chromosome (as a whole). This is because those genes benefit (selfishly) from being able to move into new genetic backgrounds (see Bergstrom et al. Genetics 2000; and especially Niehus et al. Nat. Comms. 2015). If parts of the chromosome are indeed hypermobile, theory suggests they should resemble MGEs in terms of gene content. These changes will make the authors' subsequent arguments clearer.

Corrected.

Line 43. Beware of absolutes. Plasmids can also transmit by transduction or by vesicles. Perhaps "Conjugation is the main process by which..." is better.

Corrected.

Lines 44-55 describe conjugative elements (plasmids and ICE). A couple of words might help signpost the reader at the end of this paragraph. For example, describing how conjugative machinery has commonly been held to be the dominant mechanism by which HGT is achieved. See e.g. Halary et al. 2010 (doi: 10.1073/pnas.0908978107)

Done.

Line 78. A definition of 'power' is required. In terms of rate of transduction, or material transferred, or...?

We do not think a further clarification is required here since we validate and explain this concept through all the manuscript.

Line 85. Successful horizontal gene transfer requires both transmission of DNA, and then maintenance in the recipient. For plasmids, conjugative transposons, and integrative phage, this is achieved through origins of replication and/or integrases (or recombinases/transposases, etc). For LT it is achieved through recipient-encoded recombination processes (homologous recombination, or NHEJ?). A word on this here will be useful to the reader, e.g. "...which are subsequently transferred at high frequencies (Figure 1) and can then be integrated into the recipient genome by homologous recombination.". This will help to explain the qualitative differences between LT-mediated chromosomal HGT, and MGE-mediated HGT.

Corrected.

Line 246. Typo ('by facilitated').

Deleted.

Line 276. Is there any evidence that LT-transferred DNA can integrate by NHEJ?

We have not analysed this possibility yet.

Reviewer #2 (Remarks to the Author):

The authors have adequately addressed my previous inquiries.

Thanks for the support.